

# Study of deuteron-proton backward elastic scattering at intermediate energies

**Nadezhda Ladygina** *

Joint Institute for Nuclear Research, Dubna, Russia

* nladygina@jinr.ru

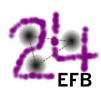

## Abstract

We study deuteron- proton elastic scattering in the deuteron energy range between 500 MeV and 2 GeV at the cms scattering angle $\theta^* \geq 140°$. The reaction is considered in the relativistic multiple scattering expansion framework. The four reaction mechanisms are included into consideration: one-nucleon exchange, single scattering, double scattering, and the term corresponding to the delta excitation in the intermediate state.

The model is applied to describe the angular dependence of the differential cross section at the deuteron energies of between 880 and 1300 MeV. Also the energy dependence of the differential cross section and polarisation observables such as tensor analyzing power $T_{20}$ and polarization transfer from the deuteron to proton $\varkappa$ are considered at the scattering angle equal to 180°. Contributions of the different reaction mechanisms into the reaction amplitude are demonstrated in comparison with the existing experimental data.

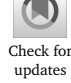

## 1 Introduction

Elastic deuteron-proton scattering is the simplest example of the hadron nucleus collision. Nowadays, a significant amount of the experimental data has been accumulated in a wide energy range both with unpolarized and polarized beams. However, we do not have any theory to describe the data for the energies above a few hundred MeV, especially, at backward scattering angles.

A good theoretical description of the deuteron-nucleon process was obtained at low energies, where the multiple scattering formalism based on the solution of the Faddeev equations, has been applied to this problem [1]. However, at the nucleon energies above 130 MeV there is some discrepancy between the experimental data and theoretical predictions in the minimum of the differential cross section [2].

The Glauber theory taking into account both single and double nucleon-nucleon interaction successfully describes the differential cross sections of the dp-elastic scattering at small angles [3]- [4]. But it does not properly work at larger scattering angles.

In 1969 A.Kerman and L.Kisslinger supposed that resonances can play an important role in deuteron-proton backward elastic scattering [5]. Later the double-scattering diagram with Δ-isobar in the intermediate state was taken into account in dp- backward scattering. The significant contribution of this term to the reaction amplitude was demonstrated in refs. [6]-[8]. However, the double scattering with nucleon in an intermediate state was not considered in these papers. Perhaps, it was the reason why the description of the differential cross sections energy dependence was not good enough.

The effort to take the Δ-isobar into account in order to describe dp-elastic scattering was also done in [9], [10]. In these papers deuteron-proton scattering was considered in a whole angular range, not only at $\theta^* = 180°$. Unfortunately, the process was studied at low energies, $T_d < 200$ MeV, where the Δ-isobar excitation effects are negligible.

We have previously proposed to use a model based on the multiple expansion of the reaction amplitude in powers of the nucleon-nucleon t-matrix [11]- [13]. Here we apply the model for description of the deuteron-proton elastic scattering in backward kinematics.

## 2 General formalism

According to the three-body collision theory, the amplitude of the deuteron-proton elastic scattering $\mathcal{J}$ is defined by the matrix element of the transition operator $U_{11}$:

$$U_{dp \to dp} = \delta(E_d + E_p - E'_d - E'_p)\mathcal{J} = \langle 1(23)|[1 - P_{12} - P_{13}]U_{11}|1(23)\rangle. \tag{1}$$

Here, the state $|1(23)\rangle$ corresponds to the configuration, when nucleons 2 and 3 form the deuteron state and nucleon 1 is free. The permutation operators for two nucleons $P_{ij}$ reflect the fact that the initial and final states are antisymmetric due to the two particles exchange.

The transition operators for rearrangement scattering are defined by the Alt–Grassberger–Sandhas equations:

$$\begin{aligned}
U_{11} &= & t_2 g_0 U_{21} + t_3 g_0 U_{31}, \\
U_{21} &= & g_0^{-1} + t_1 g_0 U_{11} + t_3 g_0 U_{31}, \\
U_{31} &= & g_0^{-1} + t_1 g_0 U_{11} + t_2 g_0 U_{21},
\end{aligned} \tag{2}$$

where $t_1 = t(2,3)$, etc., is the $t$-matrix of the two-nucleon interaction and $g_0$ is the free three-particle propagator. The indices $ij$ for the transition operators $U_{ij}$ denote free particles $i$ and $j$ in the final and initial states, respectively.

Iterating these equations up to the $t_i$-second-order terms, we can present the reaction amplitude as a sum of the four contributions:

$$\mathcal{J}_{dp \to dp} = \mathcal{J}_{\text{ONE}} + \mathcal{J}_{\text{SS}} + \mathcal{J}_{\text{DS}} + \mathcal{J}_\Delta, \tag{3}$$

one-nucleon exchange, single scattering, double scattering, and rescattering with Δ -excitation in the intermediate state.

The first term in the $dp$-elastic scattering amplitude $\mathcal{J}$ in Eq.(3) is the one nucleon exchange (ONE) term.

$$\mathcal{J}_{ONE} = -2\langle 1(23)|P_{12}g_0^{-1}|1(23)\rangle. \tag{4}$$

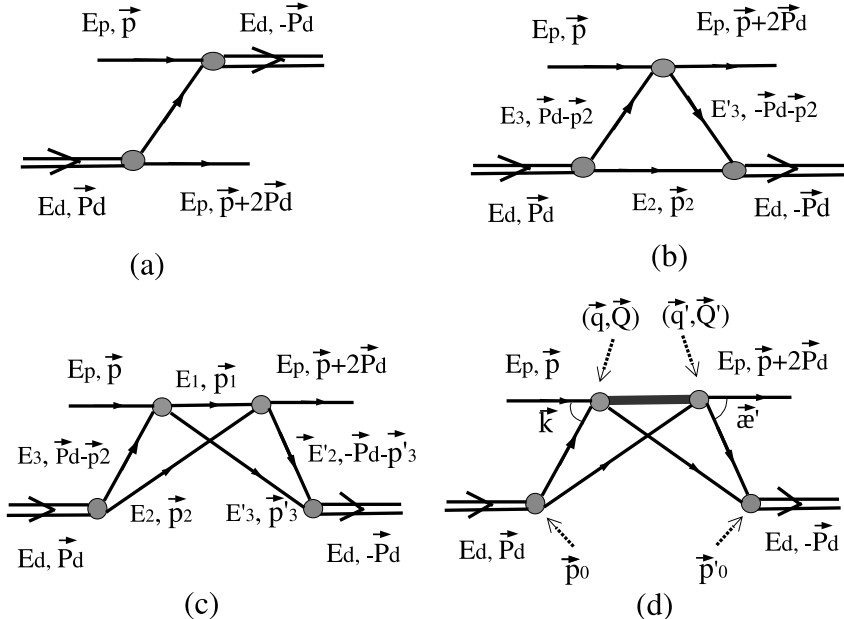

Figure 1: The diagrams included into consideration: (a) the one-nucleon exchange diagram; (b) the single scattering diagram; (c) the double scattering diagram with a nucleon in the intermediate state; (d) the double scattering diagram with $\Delta$-isobar in the intermediate state.

The corresponding diagram is presented in Fig. 1(a). Applying the definitions of the wave function of a moving deuteron and three- nucleon free propagator, we can write ONE amplitude in the following form:

$$
\begin{aligned}
\mathcal{J}_{ONE} \;=\; & -\frac{1}{2}(E_d - E_p - \sqrt{m_N^2 + \vec{p}\,^2 - \vec{P}_d{}^2}) \cdot \\
& \left\langle \vec{p}\,'m'; -\vec{P}_d \mathcal{M}'_d \big| \Omega_d^\dagger(23)[1 + (\boldsymbol{\sigma_1 \sigma_2})] \Omega_d(23) \big| \vec{P}_d \mathcal{M}_d; \vec{p}m \right\rangle,
\end{aligned}
\tag{5}
$$

where the definition of the permutation operator in spin space $P_{12}(\sigma) = \frac{1}{2}[1 + (\boldsymbol{\sigma_1 \sigma_2})]$ has been applied.

All the calculations are performed in the deuteron Breit frame, where the deuterons move in opposite directions with equal momenta (Fig. 1). It allows us to minimize the relative momenta of the nucleons in the both deuterons. As a consequence, the non-relativistic deuteron wave function can be applied in the energy range under consideration.

In the rest frame the non-relativistic wave function of the deuteron depends only on one variable $\vec{p}_0$, which is the relative momentum of the outgoing proton and neutron:

$$
\left\langle \mu_p \mu_n \big| \Omega_d \big| \mathcal{M}_d \right\rangle = \frac{1}{\sqrt{4\pi}} \left\langle \mu_p \mu_n \big| \left\{ u(p_0) + \frac{w(p_0)}{\sqrt{8}} [3(\boldsymbol{\sigma_1} \hat{p}_0)(\boldsymbol{\sigma_2} \hat{p}_0) - (\boldsymbol{\sigma_1 \sigma_2})] \right\} \big| \mathcal{M}_d \right\rangle,
$$

where $u(p_0)$ and $w(p_0)$ describe the $S$ and $D$ components of the deuteron wave function [14], [15], [16], $\hat{p}_0$ is the unit vector in $\vec{p}_0$ direction.

In order to get the wave function of the moving deuteron, it is necessary to apply the Lorenz transformations for the kinematical variables and Wigner rotations for the spin states. This procedure has been expounded in ref. [11]. The proton-neutron relative momenta for

the initial $\vec{p}_0$ and final $\vec{p}_0'$ deuterons are expressed as:

$$\vec{p}_0 = \vec{p} + \vec{P}_d\left[1 + \frac{E_n + E^*}{E_p + E_n + E^*}\right], \qquad \vec{p}_0' = \vec{p} + \vec{P}_d\left[1 - \frac{E_n + E^*}{E_p + E_n + E^*}\right]. \tag{6}$$

Here $E_n = \sqrt{m_N^2 + \vec{p}^{\;2} - \vec{P}_d^{\;2}}$ and $E^* = \sqrt{(E_p + E_n)^2 - \vec{P}_d^{\;2}}/2$ are the struck neutron energy in the moving deuteron frame and rest deuteron frame, respectively. Note, that $|\vec{p}_0| = |\vec{p}_0'|$.

The next term in the $dp$-elastic scattering amplitude Eq.(3) is the single scattering one.

$$\mathcal{J}_{SS} = 2\langle 1(23)|[1 - P_{12}]t_3|1(23)\rangle. \tag{7}$$

The corresponding diagram is presented in Fig. 1(b). Following the standard procedure we get the expression for the single scattering amplitude:

$$\mathcal{J}_{SS} = \int d\vec{q}\,'\langle -\vec{P}_d\mathcal{M}_d'|\Omega_d^\dagger|\vec{q}\,'m'', -\vec{P}_d - \vec{q}\,'m_3'\rangle \tag{8}$$
$$\langle \vec{p}\,'m', -\vec{P}_d - \vec{q}\,'|\frac{3}{2}t_{12}^1 + \frac{1}{2}t_{12}^0|\vec{p}m, \vec{P}_d - \vec{q}\,'m_2'\rangle \langle \vec{q}\,'m'', \vec{P}_d - \vec{q}\,'m_2'|\Omega_d|\vec{P}_d\mathcal{M}_d\rangle.$$

The relative momenta of two nucleons for the initial and final deuterons are

$$\vec{p}_0 = \vec{q}\,' - \vec{P}_d\frac{E_2 + E^*}{E_2 + E_3 + 2E^*} \quad \vec{p}_0' = \vec{q}\,' + \vec{P}_d\frac{E_2 + E'^*}{E_2 + E_3' + 2E'^*}, \tag{9}$$

where the nucleons energies $E_2$, $E_3$, $E_3'$ in the reference frame are defined by the standard manner (Fig.1b)

$$E_2 = \sqrt{m_N^2 + \vec{q}\,'^2}, \qquad E_3 = \sqrt{m_N^2 + (\vec{P}_d - \vec{q}\,')^2}, \qquad E_3' = \sqrt{m_N^2 + (\vec{P}_d + \vec{q}\,')^2} \tag{10}$$

and these energies in the center-of-mass of the two nucleons forming the initial and final deuterons are equal, correspondingly, to

$$E^* = \frac{1}{2}\sqrt{(E_2 + E_3)^2 - \vec{P}_d^2}, \qquad E'^* = \frac{1}{2}\sqrt{(E_2 + E_3')^2 - \vec{P}_d^2}. \tag{11}$$

The nucleon-nucleon scattering is described by the t-matrix $t_{ij}^T$. We use the parameterization of this matrix offered by Love and Franey [17]. This is the on-shell NN t-matrix defined in the center-of-mass:

$$\langle \varkappa^{*\prime}\mu_1'\mu_2'|t_{c.m.}|\varkappa^*\mu_1\mu_2\rangle = \langle \varkappa^{*\prime}\mu_1'\mu_2'|A + B(\sigma_1\hat{N}^*)(\sigma_2\hat{N}^*) + \tag{12}$$
$$C(\sigma_1 + \sigma_2)\cdot\hat{N}^* + D(\sigma_1\hat{q}^*)(\sigma_2\hat{q}^*) + F(\sigma_1\hat{Q}^*)(\sigma_2\hat{Q}^*)|\varkappa^*\mu_1\mu_2\rangle.$$

The orthonormal basis $\{\hat{q}^*, \hat{Q}^*, \hat{N}^*\}$ is a combination of the nucleon relative momenta in the initial $\varkappa^*$ and final $\varkappa^{\prime *}$ states:

$$\hat{q}^* = \frac{\varkappa^* - \varkappa^{*\prime}}{|\varkappa^* - \varkappa^{*\prime}|}, \quad \hat{Q}^* = \frac{\varkappa^* + \varkappa^{*\prime}}{|\varkappa^* + \varkappa^{*\prime}|}, \quad \hat{N}^* = \frac{\varkappa^* \times \varkappa^{*\prime}}{|\varkappa^* \times \varkappa^{*\prime}|}. \tag{13}$$

The amplitudes $A, B, C, D, F$ are the functions of the center-of-mass energy and scattering angle. The radial parts of these amplitudes are taken as a sum of Yukawa terms. A new fit of the model parameters [18] was done in accordance with the phase-shift-analysis data SP07 [19].

Since the matrix elements are expressed via the effective $NN$-interaction operators sandwiched between the initial and final plane-wave states, this construction can be extended to the off-shell case allowing the initial and final states to get the current values of $\varkappa$ and $\varkappa'$. Obviously, this extrapolation does not change the general spin structure.

The double scattering contribution (Fig.1c) is defined by a deuteron wave function and two nucleon-nucleon t-matrixes. Also we have here three-nucleon propagator:

$$\mathcal{J}_{DS} = \int d\vec{p}_2 d\vec{p}_3' \left\langle -\vec{P}_d \mathcal{M}_d' \middle| \Omega_d^\dagger \middle| -\vec{P}_d - \vec{p}_3' \ m_2', \vec{p}_3' \ m_3' \right\rangle \tag{14}$$

$$\left\langle \vec{p}' \ m', -\vec{P}_d - \vec{p}_3' \ m_2', \vec{p}_3' \ m_3' \middle| \right.$$

$$\left( \frac{t_{3(NN)}^1(E') t_{2(NN)}^1(E) + [t_{3(NN)}^1(E') + t_{3(NN)}^0(E')][t_{2(NN)}^1(E) + t_{2(NN)}^0(E)]/4}{E_d + E_p - E_1 - E_2 - E_3' + i\varepsilon} \right)$$

$$\left. \middle| \vec{p} \ m, \vec{p}_2 \ m_2, \vec{P}_d - \vec{p}_2 \ m_3 \right\rangle \left\langle \vec{p}_2 \ m_2, \ \vec{P}_d - \vec{p}_2 \ m_3 \middle| \Omega_d \middle| \vec{P}_d \mathcal{M}_d \right\rangle.$$

The argument of the $NN$-matrix is defined as the three-nucleon on-shell energy excluding the energy of the nucleon which does not participate in the interaction:

$$E = E_d + E_p - E_2, \qquad E' = E_d + E_p - E_3'. \tag{15}$$

The structure of the delta amplitude (Fig.1d) looks like the double-scattering one. But here we have $NN \rightarrow \Delta N$ matrices instead the nucleon-nucleon matrixes and $NN\Delta$-propagator instead three-nucleon one.

$$\mathcal{J}_\Delta = 2 \int d\vec{p}_2 d\vec{p}_3' dE_\Delta d\vec{p}_\Delta \delta\left( E_\Delta - \sqrt{\mu^2 + \vec{p}_\Delta^2} \right) \delta(\vec{p} + \vec{P}_d - \vec{p}_2 - \vec{p}_3 - \vec{p}_\Delta) {}_1\left\langle \frac{1}{2}\tau' \frac{1}{2}m'\vec{p}' \middle| \right.$$

$${}_{23}\left\langle 00; -\vec{P}_d 1 \mathcal{M}_d' \middle| \Omega_d^\dagger [1 - P_{12}] t_{3(N\Delta)}(E') \frac{1}{E - E_2 - E_3' - E_\Delta + i\Gamma(E_\Delta/2)} \middle| \Psi_{\vec{p}_\Delta}(E_\Delta) \right\rangle_1$$

$$\left| \frac{1}{2}\tau_2 \frac{1}{2}m_2\vec{p}_2; \frac{1}{2}\tau_3 \frac{1}{2}m_3\vec{p}_3 \right\rangle_{23} {}_{23}\left\langle \frac{1}{2}\tau_2 \frac{1}{2}m_2\vec{p}_2; \frac{1}{2}\tau_3 \frac{1}{2}m_3\vec{p}_3 \middle| \right. \tag{16}$$

$${}_1\left\langle \Psi_{\vec{p}_\Delta}(E_\Delta) \middle| t_{2(N\Delta)}(E)[1 - P_{13}]\Omega_d \middle| \vec{P}_d 1\mathcal{M}_d; 00 \right\rangle_{23} \left| \frac{1}{2}\tau \frac{1}{2}m\vec{p} \right\rangle_1.$$

Here a full set of the particles quantum numbers was included into the amplitude definition. Isospin and spin quantum numbers are marked by $\tau$ and $m$ or $M$, respectively. The indexes near the bracket correspond to the particles numbers.

The distribution function of the delta energy

$$\left| \Psi_{\vec{p}_\Delta}(E_\Delta) \right\rangle\!\!\left\langle \Psi_{\vec{p}_\Delta}(E_\Delta) \right| = \rho(E_\Delta) \tag{17}$$

is defined through the delta width $\Gamma(\mu)$:

$$\rho(\mu) = \frac{1}{2\pi} \frac{\Gamma(\mu)}{(E_\Delta(\mu) - E_\Delta(m_\Delta))^2 + \Gamma^2(\mu)/4}, \tag{18}$$

where $\mu^2 = E_\Delta^2 - \vec{p}_\Delta^2$ is the squared four-momentum of the delta. The delta width is energy dependent. We use, here, the standart parameterization of $\Gamma(\mu)$ taking into account the $\Delta$ off-shell corrections:

$$\Gamma(\mu) = \Gamma_0 \frac{p^3(\mu^2, m_\pi^2)}{p^3(m_\Delta^2, m_\pi^2)} \cdot \frac{p^2(m_\Delta^2, m_\pi^2) + \gamma^2}{p^2(\mu^2, m_\pi^2) + \gamma^2}, \tag{19}$$

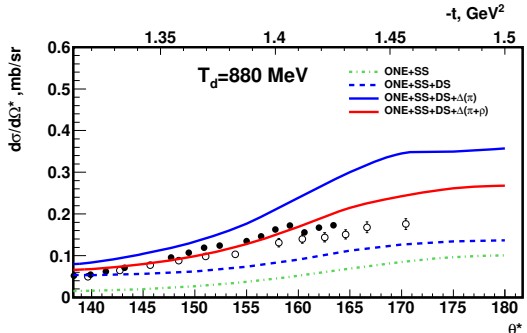

Figure 2: The angular dependence of the differential cross section at the deuteron energy $T_d = 880$ MeV. The data are from ∘ - [22] at $T_d = 850$ MeV •- [23] at $T_d = 940$ MeV

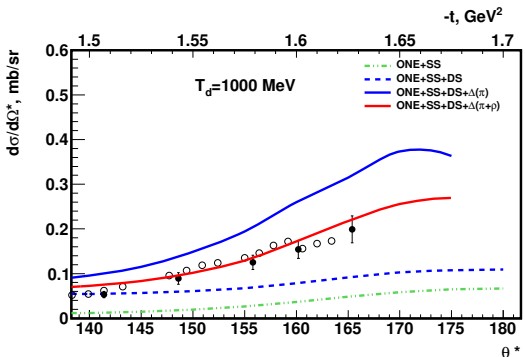

Figure 3: The angular dependence of the differential cross section at the deuteron energy at $T_d = 1000$ MeV. The data are from ∘- [22] at $T_d = 940$ MeV, •- [24] at $T_d = 1169$ MeV.

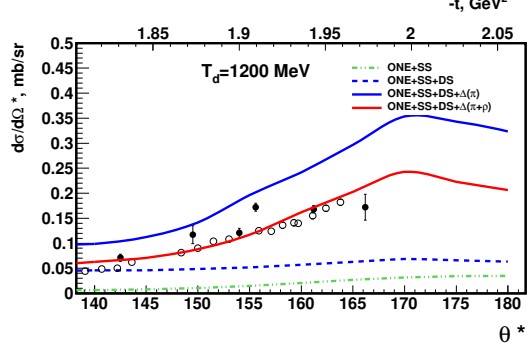

Figure 4: The angular dependence of the differential cross section at the deuteron energy at $T_d = 1200$ MeV. The data are from ∘ - [22] at $T_d = 1169$ MeV, •- [25] at $T_d = 1200$ MeV.

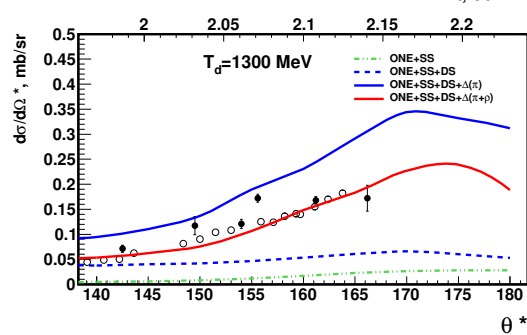

Figure 5: The angular dependence of the differential cross section at the deuteron energy $T_d = 1300$ MeV. The data are from ∘ - [22] at $T_d = 1169$ MeV, •- [25] at $T_d = 1200$ MeV.

where $p(x^2, m_\pi^2)$ is the momentum in the $\pi N$ -centere-of-mass:

$$p(x^2, m_\pi^2) = \sqrt{(x^2 + m_N^2 - m_\pi^2)^2/4x^2 - m_N^2}. \tag{20}$$

In our calculation we use the following value:

$$\Gamma_0 = 0.120 \; GeV, \quad \gamma = 0.200 \; GeV, \quad m_\Delta = 1.232. \tag{21}$$

In the Born approximation the $NN \to N\Delta$ t-matrix can be replaced with the corresponding potential:

$$\left\langle \vec{p}, \frac{1}{2}m, \frac{1}{2}\tau \middle| t_{(N\Delta)}(E) \middle| \Psi_{\vec{p}_\Delta}(E_\Delta) \right\rangle \approx \left\langle \vec{p}, \frac{1}{2}m, \frac{1}{2}\tau \middle| V_{(N\Delta)}(E) \middle| \Psi_{\vec{p}_\Delta}(E_\Delta) \right\rangle. \tag{22}$$

The potential for the $NN \to N\Delta$ transition is based on the $\pi-$ and $\rho-$ exchanges:

$$V_{\beta\alpha}^{(\pi)} = -\frac{f_\pi f_\pi^*}{m_\pi^2} F_\pi^2(t) \frac{q^2}{m_\pi^2 - t} (\vec{\sigma} \cdot \hat{q})(\vec{S} \cdot \hat{q})(\vec{\tau} \cdot \vec{T}) \tag{23}$$

$$V_{\beta\alpha}^{(\rho)} = -\frac{f_\rho f_\rho^*}{m_\rho^2} F_\rho^2(t) \frac{q^2}{m_\rho^2 - t} \{(\vec{\sigma}\vec{S}) - (\vec{\sigma} \cdot \hat{q})(\vec{S} \cdot \hat{q})\}(\vec{\tau} \cdot \vec{T}).$$

Here, $t$ is the four transfer momentum and $\vec{q}$ is the corresponding three transfer momentum. The operators $\vec{\sigma}(\vec{\tau})$ are $\frac{1}{2}$- spin (isospin) operators defined by Pauli matrixes while $\vec{S}(\vec{T})$

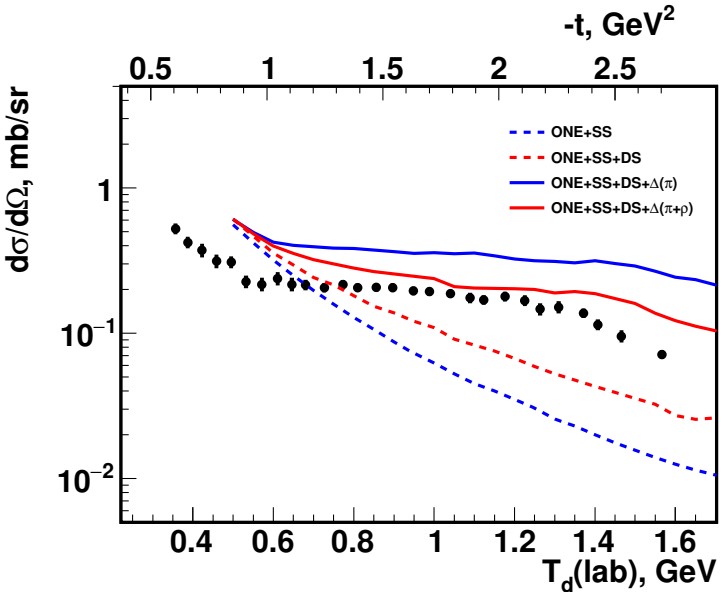

Figure 6: The energy dependence of the differential cross section at the scattering angle $\theta^* = 180°$. The data are from [26].

operators correspond to $\frac{1}{2} \to \frac{3}{2}$ spin (isospin) transition. $m_\pi$ and $m_\rho$ are a pion and $\rho$- meson-masses. The coupling constant $f_\pi$ is related with the $NN\pi$ vertex and $f_\pi^*$ corresponds to the $N\Delta\pi$ one. It concerns also $\rho-$ coupling constants.

$$
\begin{aligned}
f_\pi &= 1.008 & f_\pi^* &= 2.156 \\
f_\rho &= 7.8 & f_\rho^* &= 1.85 f_\rho.
\end{aligned}
\tag{24}
$$

The hadronic form factor was chosen in a pole form:

$$
F_x(t) = \left[ (\Lambda_x^2 - m_x^2)/(\Lambda_x^2 - t) \right]^n .
\tag{25}
$$

In our calculation we use $\Lambda_\pi = 0.8$ GeV, $\Lambda_\rho = 1.8$ GeV. The exponent $n$ is equal to 1 for $\pi$-meson and 2 for $\rho$-meson.

Since two nucleon states in the $NN \to N\Delta$ vertexes are antisymmetrized, two permutation operators appear in Eq.(16). As consequence, the $\Delta$- amplitude contains four terms: one direct, two exchange, and one double-exchange ones. The permutation operator $P_{ij}$ involves the permutation of all quantum numbers. Here, it is permutation over momentum, spin, and isospin indexes: $P_{ij} = P_{ij}(p) P_{ij}(\sigma) P_{ij}(\tau)$.

## 3 Results

We applied the method to describe angular dependences of the differential cross sections at the backward scattering angles $\theta^* \geq 140°$ at four deuteron energies of 880, 1000, 1200, and



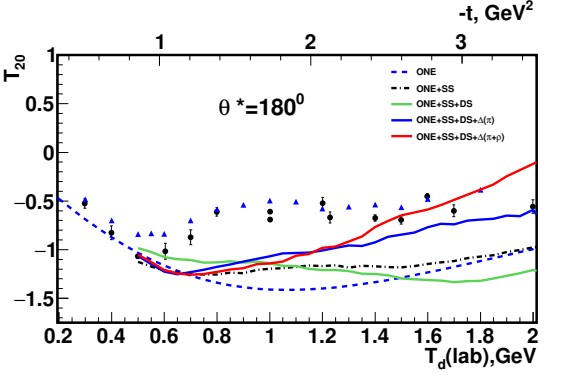

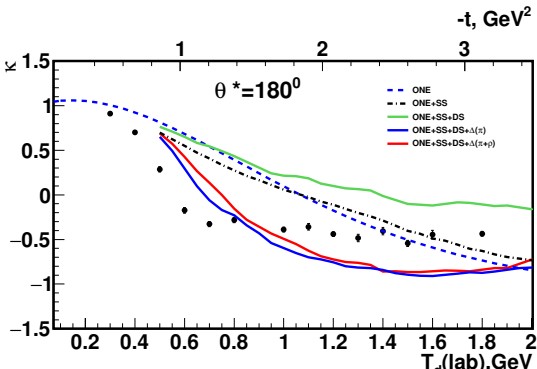

Figure 7: The energy dependence of the tensor analyzing power $T_{20}$ at the scattering angle $\theta^* = 180°$. The data are taken from ● - [27], ▲ - [28].

Figure 8: The energy dependence of the polarisation transfer at the scattering angle $\theta^* = 180°$. The data are taken from [28].

1300 MeV. In Figs.2-5 the calculation results are presented for four cases. The dashed-dotted curves correspond to the results obtained taking into account only ONE and single-scattering contributions. The results obtained with the addition of the double-scattering term are represented by dashed curves. The solid curves correspond to the theory predictions taking into account also $\Delta$-isobar in the intermediate state with both $\pi$-meson (blue line) and $\pi$- and $\rho$-mesons (red line).

It is well known that the data on the differential cross sections show some enhancement at the backward angles. However the calculation results obtained without the $\Delta$-isobar lie below the data. Moreover the difference between the data and the results increases with the energy growing. When we include the $\Delta$-isobar into consideration we get rather good agreement between the data and theory in the case when $\pi$- and $\rho$-mesons are taken into account. When we include only $\pi$-meson in the $\Delta$-isobar description we get overestimated values for the differential cross sections.

It is interesting to look at a manifestation of the various mechanisms at the critical scattering angle of 180°. An energy dependence of the differential cross section is presented in Fig. 6. The data demonstrate a shoulder at the energies between about 500 and 1400 MeV which is not described by the calculations without $\Delta$-isobar. The curves obtained taking into account only ONE+SS and ONE+SS+DS rapidly descend and pass below the data. Inclusion $\Delta$-isobar into consideration allows us to describe the shoulder and significantly improve an agreement between the data and theoretical predictions. As in the case of the angular distributions of the differential cross sections, we get overestimated values when we include only $\pi$- meson in the $\Delta$-isobar definition.

Tensor analyzing power $T_{20}$ is presented in Fig. 7 as a function of the deuteron energy. The result of the simplest reaction mechanism ONE is close to the data at low energies up to about 500 MeV. But then the data go up while ONE curve descends up to minimum equal to $-\sqrt{2}$ at the deuteron energy of about 1 GeV. The addition of the single- and double- scattering terms allows to slightly rise the curves in the minimum but the data description remains unsatisfactory. The results obtained with $\Delta$-isobar are closer to the data except for the energy range between 700 and 1100 MeV where the data show the rise.

The role of the $\Delta$-isobar is clearly manifested in the polarization transfer $\varkappa$, which is shown

in Fig. 8 versus the deuteron energy. The data show the precipitous fall at the deuteron energy between 400 and 600 MeV and then go on a plateau. The results obtained with the inclusion of $\Delta$-isobar reproduce the shape of the data while the results of the calculations performed without $\Delta$-isobar are far from the data.

## 4  Conclusion

We considered dp- backward elastic scattering taking into account four contributions: one-nucleon-exchange, single-scattering, double-scattering, and $\Delta$-excitation in an intermediate state. We showed a role of each reaction mechanism in the description of the angular dependence of the differential cross section. Inclusion of the $\Delta$-isobar term into consideration allowed to describe the enhancement of the differential cross section at $\theta^* \geq 140°$ in the energy range between 880 and 1300 MeV.

The reaction mechanisms were also studied at the scattering angle $\theta^* = 180°$. It was obtained a quite good agreement between the experimental data and the theoretical predictions for the energy dependence of the differential cross section. Some progress was achieved in the description of the tensor analyzing power $T_{20}$ and polarisation transfer $\kappa$.

## Acknowledgements

The author is grateful to Dr. V.P. Ladygin for fruitful discussions and interest in this problem. This work has been supported by the Russian Foundation for Basic Research under grant №19-02-00079a.

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
