# Peer review of "Study of deuteron-proton backward elastic scattering at intermediate energies."

_SciPost Physics Proceedings, doi:SciPost Phys. Proc. 3, 053 (2020)_

## Round 1 · Referee Report · Paul Stevenson (Referee 1) · 2019-12-17

Report

The paper describes theoretical results of multiple scattering theory for backwards scattering at high energy of dp reactions. A term-by-term analysis is made to understand which terms (diagrams) contribute most significantly to the cross sections. The results are interesting and show in general that inclusion of all terms considered (i.e. including intermediate states) is necessary to get good agreement with experiment. I recommend the paper for publication, as a good summary of the work presented at the conference Small correction: before equation (6) I believe it should be the Lorentz (note spelling) transformations for the kinematic variables. This can be corrected in the proof stage.

---

## Editorial Decision

published